# Exploring Active 3D Object Detection from a Generalization Perspective

**Yadan Luo**[*]**, Zhuoxiao Chen**[*]**, Zijian Wang, Xin Yu, Zi Huang, Mahsa Baktashmotlagh**
The University of Queensland, Australia

## Abstract

To alleviate the high annotation cost in LiDAR-based 3D object detection, active learning is a promising solution that learns to select only a small portion of unlabeled data to annotate, without compromising model performance. Our empirical study, however, suggests that mainstream uncertainty-based and diversity-based active learning policies are not effective when applied in the 3D detection task, as they fail to balance the trade-off between point cloud informativeness and box-level annotation costs. To overcome this limitation, we jointly investigate three novel criteria in our framework **CRB** for point cloud acquisition - *label conciseness*, *feature representativeness* and *geometric balance*, which hierarchically filters out the point clouds of redundant 3D bounding box labels, latent features and geometric characteristics (*e.g.*, point cloud density) from the unlabeled sample pool and greedily selects informative ones with fewer objects to annotate. Our theoretical analysis demonstrates that the proposed criteria aligns the marginal distributions of the selected subset and the prior distributions of the unseen test set, and minimizes the upper bound of the generalization error. To validate the effectiveness and applicability of CRB, we conduct extensive experiments on the two benchmark 3D object detection datasets of KITTI and Waymo and examine both one-stage (*i.e.*, SECOND) and two-stage 3D detectors (*i.e.*, Pv-RCNN). Experiments evidence that the proposed approach outperforms existing active learning strategies and achieves fully supervised performance requiring $1\%$ and $8\%$ annotations of bounding boxes and point clouds, respectively. Source code: https://github.com/Luoyadan/CRB-active-3Ddet.

## 1 Introduction

LiDAR-based 3D object detection plays an indispensable role in 3D scene understanding with a wide range of applications such as autonomous driving (Deng et al., 2021; Wang et al., 2020) and robotics (Ahmed et al., 2018; Montes et al., 2020; Wang et al., 2019). The emerging stream of 3D detection models enables accurate recognition at the cost of large-scale labeled point clouds, where 7-degree of freedom (DOF) 3D bounding boxes - consisting of a position, size, and orientation information- for each object are annotated. In the benchmark datasets like Waymo (Sun et al., 2020), there are over 12 million LiDAR boxes, for which, labeling a precise 3D box takes more than 100 seconds for an annotator (Song et al., 2015). This prerequisite for the performance boost greatly hinders the feasibility of applying models to the wild, especially when the annotation budget is limited.

To alleviate this limitation, active learning (AL) aims to reduce labeling costs by querying labels for only a small portion of unlabeled data. The criterion-based query selection process iteratively selects the most beneficial samples for the subsequent model training until the labeling budget is run out. The criterion is expected to quantify the sample informativeness using the heuristics derived from *sample uncertainty* (Gal et al., 2017; Du et al., 2021; Caramalau et al., 2021; Yuan et al., 2021; Choi et al., 2021; Zhang et al., 2020; Shi & Li, 2019) and *sample diversity* (Ma et al., 2021; Gudovskiy et al., 2020; Gao et al., 2020; Sinha et al., 2019; Pinsler et al., 2019). In particular, uncertainty-driven approaches focus on the samples that the model is the least confident of their labels, thus searching for the candidates with: maximum entropy (MacKay, 1992; Shannon, 1948; Kim et al., 2021b; Siddiqui et al., 2020; Shi & Yu, 2019), disagreement among different experts (Freund et al., 1992; Tran et al., 2019), minimum posterior probability of a predicted class (Wang et al., 2017), or the samples

---
[*]Equal contribution. Correspondence to Yadan Luo <y.luo@uq.edu.au>.

with reducible yet maximum estimated error (Roy & McCallum, 2001; Yoo & Kweon, 2019; Kim et al., 2021a). On the other hand, diversity-based methods try to find the most representative samples to avoid sample redundancy. To this end, they form subsets that are sufficiently diverse to describe the entire data pool by making use of the greedy coreset algorithms (Sener & Savarese, 2018), or the clustering algorithms (Nguyen & Smeulders, 2004). Recent works (Liu et al., 2021; Citovsky et al., 2021; Kirsch et al., 2019; Houlsby et al., 2011) combine the aforementioned heuristics: they measure uncertainty as the gradient magnitude of samples (Ash et al., 2020) or its second-order metrics (Liu et al., 2021) at the final layer of neural networks, and then select samples with gradients spanning a diverse set of directions. While effective, the hybrid approaches commonly cause heavy computational overhead, since gradient computation is required for each sample in the unlabeled pool. Another stream of works apply active learning to 2D/3D object detection tasks (Feng et al., 2019; Schmidt et al., 2020; Wang et al., 2022; Wu et al., 2022; Tang et al., 2021), by leveraging ensemble (Beluch et al., 2018) or Monte Carlo (MC) dropout (Gal & Ghahramani, 2016) algorithms to estimate the classification and localization uncertainty of bounding boxes for images/point clouds acquisition (more details in Appendix I). Nevertheless, those AL methods generally favor the point clouds with more objects, which have a higher chance of containing uncertain and diverse objects. With a fixed annotation budget, it is far from optimal to select such point clouds, since more clicks are required to form 3D box annotations.

To overcome the above limitations, we propose to learn AL criteria for cost-efficient sample acquisition at the 3D box level by empirically studying its relationship with optimizing the generalization upper bound. Specifically, we propose three selection criteria for cost-effective point cloud acquisition, termed as CRB, *i.e., label conciseness*, *feature representativeness* and *geometric balance*. Specifically, we divide the sample selection process into three stages: (1) To alleviate the issues of label redundancy and class imbalance, and to ensure *label conciseness*, we firstly calculate the entropy of bounding box label predictions and only pick top $\mathcal{K}_1$ point clouds for Stage 2; (2) We then examine the *feature representativeness* of candidates by formulating the task as the $\mathcal{K}_2$-medoids problem on the gradient space. To jointly consider the impact of classification and regression objectives on gradients, we enable the Monte Carlo dropout (MC-DROPOUT) and construct the hypothetical labels by averaging predictions from multiple stochastic forward passes. (3) Finally, to maintain the *geometric balance* property, we minimize the KL divergence between the marginal distributions of point cloud density of each predicted bounding box. This makes the trained detector predict more accurate localization and size of objects, and recognize both close (*i.e.*, dense) and distant (*i.e.*, sparse) objects at the test time, using minimum number of annotations. We base our criterion design on our theoretical analysis of optimizing the upper bound of the generalization risk, which can be reformulated as distribution alignment of the selected subset and the test set. Note that since the empirical distribution of the test set is not observable during training, WLOG, we make an appropriate assumption of its prior distribution.

**Contributions**. Our work is a pioneering study in active learning for 3D object detection, aiming to boost the detection performance at the **lowest cost of bounding box-level annotations**. To this end, we propose a hierarchical active learning scheme for 3D object detection, which progressively filters candidates according to the derived selection criteria without triggering heavy computation. Extensive experiments conducted demonstrate that the proposed CRB strategy can consistently outperform all the state-of-the-art AL baselines on two large-scale 3D detection datasets irrespective of the detector architecture. To enhance the reproducibility of our work and accelerate future work in this new research direction, we develop an `active-3D-det` toolbox, which accommodates various AL approaches and 3D detectors.

## 2 METHODOLOGY

### 2.1 PROBLEM FORMULATION

In this section, we mathematically formulate the problem of active learning for 3D object detection and set up the notations. Given an orderless LiDAR point cloud $\mathcal{P} = \{x, y, z, e\}$ with 3D location $(x, y, z)$ and reflectance $e$, the goal of 3D object detection is to localize the objects of interest as a set of 3D bounding boxes $\mathcal{B} = \{b_k\}_{k \in [N_B]}$ with $N_B$ indicating the number of detected bounding boxes, and predict the associated box labels $Y = \{y_k\}_{k \in [N_B]} \in \mathcal{Y} = \{1, \ldots, C\}$, with $C$ being the number of classes to predict. Each bounding box $b$ represents the relative center position $(p_x, p_y, p_z)$ to the object ground planes, the box size $(l, w, h)$, and the heading angle $\theta$. Mainstream 3D object detectors use point clouds $\mathcal{P}$ to extract point-level features $\boldsymbol{x} \in \mathbb{R}^{W \cdot L \cdot F}$ (Shi et al., 2019; Yang et al., 2019;

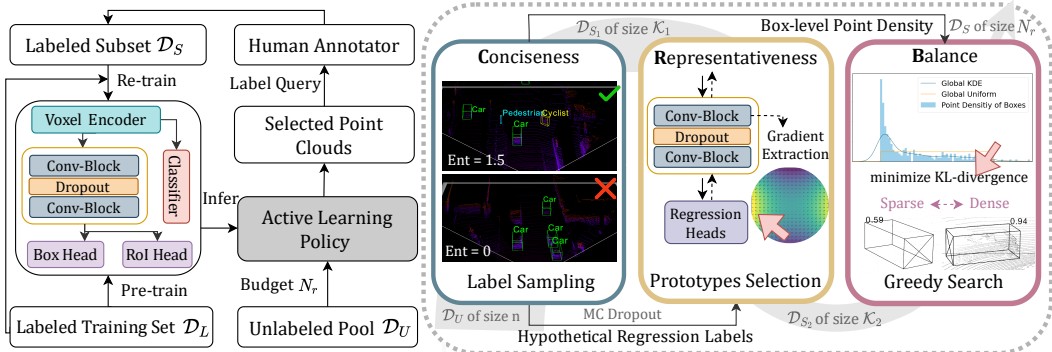

Figure 1: An illustrative flowchart of the proposed CRB framework for active selection of point clouds. Motivated by optimizing the generalization risk, the derived strategy hierarchically selects point clouds that have non-redundant bounding box labels, latent gradients and geometric characteristics to mitigate the gap with the test set and minimize annotation costs.

2020) or by voxelization (Shi et al., 2020), with $W$, $L$, $F$ representing width, length, and channels of the feature map. The feature map $x$ is passed to a classifier $f(\cdot; w_f)$ parameterized by $w_f$ and regression heads $g(\cdot; w_g)$ (*e.g.*, box refinement and ROI regression) parameterized by $w_g$. The output of the model is the detected bounding boxes $\widehat{\mathcal{B}} = \{\hat{b}_k\}$ with the associated box labels $\widehat{Y} = \{\hat{y}_k\}$ from anchored areas. The loss functions $\ell^{cls}$ and $\ell^{reg}$ for classification (*e.g.*, regularized cross entropy loss Oberman & Calder (2018)) and regression (*e.g.*, mean absolute error/$L_1$ regularization Qi et al. (2020)) are assumed to be Lipschitz continuous. As shown in the left half of Figure 1, in an active learning pipeline, a small set of labeled point clouds $\mathcal{D}_L = \{(\mathcal{P}, \mathcal{B}, Y)_i\}_{i \in [m]}$ and a large pool of raw point clouds $\mathcal{D}_U = \{(\mathcal{P})_j\}_{j \in [n]}$ are provided at training time, with $n$ and $m$ being a total number of point clouds and $m \ll n$. For each active learning round $r \in [R]$, and based on the criterion defined by an active learning policy, we select a subset of raw data $\{\mathcal{P}_j\}_{j \in [N_r]}$ from $\mathcal{D}_U$ and query the labels of 3D bounding boxes from an oracle $\Omega : \mathcal{P} \to \mathcal{B} \times \mathcal{Y}$ to construct $\mathcal{D}_S = \{(\mathcal{P}, \mathcal{B}, Y)_j\}_{j \in [N_r]}$. The 3D detection model is pre-trained with $\mathcal{D}_L$ for active selection, and then retrained with $\mathcal{D}_S \cup \mathcal{D}_L$ until the selected samples reach the final budget $B$, *i.e.*, $\sum_{r=1}^{R} N_r = B$.

## 2.2 THEORETICAL MOTIVATION

The core question of active 3D detection is how to design a proper criterion, based on which a fixed number of unlabeled point clouds can be selected to achieve minimum empirical risk $\mathfrak{R}_T[\ell(f, g; w)]$ on the test set $\mathcal{D}_T$ and minimum annotation time. Below, inspired by (Mansour et al., 2009; Ben-David et al., 2010), we derive the following **generalization bound** for active 3D detection so that the desired acquisition criteria can be obtained by optimizing the generalization risk.

**Theorem 2.1.** *Let $\mathcal{H}$ be a hypothesis space of Vapnik-Chervonenkis (VC) dimension $d$, with $f$ and $g$ being the classification and regression branches, respectively. The $\widehat{\mathcal{D}}_S$ and $\widehat{\mathcal{D}}_T$ represent the empirical distribution induced by samples drawn from the acquired subset $\mathcal{D}_S$ and the test set $\mathcal{D}_T$, and $\ell$ the loss function bounded by $\mathcal{J}$. It is proven that $\forall \, \delta \in (0, 1)$, and $\forall f, g \in \mathcal{H}$, with probability at least $1 - \delta$ the following inequality holds,*

$$\mathfrak{R}_T[\ell(f, g; w)] \leq \mathfrak{R}_S[\ell(f, g; w)] + \frac{1}{2} disc(\widehat{\mathcal{D}}_S, \widehat{\mathcal{D}}_T) + \lambda^* + const,$$

*where $const = 3\mathcal{J}(\sqrt{\frac{\log \frac{4}{\delta}}{2N_r}} + \sqrt{\frac{\log \frac{4}{\delta}}{2N_t}}) + \sqrt{\frac{2d \log(eN_r/d)}{N_r}} + \sqrt{\frac{2d \log(eN_t/d)}{N_t}}$.*

*Notably, $\lambda^* = \mathfrak{R}_T[\ell(f^*, g^*; w^*)] + \mathfrak{R}_S[\ell(f^*, g^*; w^*)]$ denotes the joint risk of the optimal hypothesis $f^*$ and $g^*$, with $w^*$ being the model weights. $N_r$ and $N_t$ indicate the number of samples in the $\mathcal{D}_S$ and $\mathcal{D}_T$. The proof can be found in the supplementary material.*

**Remark.** *The first term indicates the training error on the selected subsets, which is assumed to be trivial based on the zero training assumption (Sener & Savarese, 2018). To obtain a tight upper bound of the generalization risk, the **optimal subset** $\mathcal{D}_S^*$ can be determined via minimizing the discrepancy distance of empirical distribution of two sets, i.e.,*

$$\mathcal{D}_S^* = \underset{\mathcal{D}_S \subset \mathcal{D}_U}{\arg\min} \, disc(\widehat{\mathcal{D}}_S, \widehat{\mathcal{D}}_T).$$

*Below, we define the discrepancy distance for the 3D object detection task.*

**Definition 1.** *For any $f, g, f', g' \in \mathcal{H}$, the discrepancy between the distribution of the selected sets $\mathcal{D}_S$ and unlabeled pool $\mathcal{D}_T$ can be formulated as,*

$$disc(\widehat{\mathcal{D}}_S, \widehat{\mathcal{D}}_T) = \sup_{f, f' \in \mathcal{H}} |\mathbb{E}_{\widehat{\mathcal{D}}_S} \ell(f, f') - \mathbb{E}_{\widehat{\mathcal{D}}_T} \ell(f, f')| + \sup_{g, g' \in \mathcal{H}} |\mathbb{E}_{\widehat{\mathcal{D}}_S} \ell(g, g') - \mathbb{E}_{\widehat{\mathcal{D}}_T} \ell(g, g')|,$$

*where the bounded expected loss $\ell$ for any classification and regression functions are symmetric and satisfy the triangle inequality.*

**Remark.** *As 3D object detection is naturally an integration of classification and regression tasks, mitigating the set discrepancy is basically aligning the inputs and outputs of each branch. Therefore, with the detector freezed during the active selection, finding an optimal $\mathcal{D}_S^*$ can be interpreted as enhancing the acquired set's (1) **Label Conciseness**: aligning marginal label distribution of bounding boxes, (2) **Feature Representativeness**: aligning marginal distribution of the latent representations of point clouds, and (3) **Geometric Balance**: aligning marginal distribution of geometric characteristics of point clouds and predicted bounding boxes, and can be written as:*

$$\mathcal{D}_S^* \approx \underset{\mathcal{D}_S \subset \mathcal{D}_U}{\arg\min} \overbrace{d_{\mathcal{A}}(P_{\widehat{Y}_S}, P_{Y_T})}^{Conciseness} + \overbrace{d_{\mathcal{A}}(P_{X_S}, P_{X_T})}^{Representativeness} + \overbrace{d_{\mathcal{A}}(P_{\phi(\mathcal{P}_S, \widehat{\mathcal{B}}_S)}, P_{\phi(\mathcal{P}_T, \mathcal{B}_T)})}^{Balance}. \tag{1}$$

*Here, $\mathcal{P}_S$ and $\mathcal{P}_T$ represent the point clouds in the selected set and the ones in the test set. $\phi(\cdot)$ indicates the geometric descriptor of point clouds and $d_{\mathcal{A}}$ distance (Kifer et al., 2004) which can be estimated by a finite set of samples. For latent features $X_S$ and $X_T$, we only focus on the features that differ from the training sets, since $\mathbb{E}_{\widehat{D}_L} \ell^{cls} = 0$ and $\mathbb{E}_{\widehat{D}_L} \ell^{reg} = 0$ based on the zero training error assumption. Considering that test samples and their associated labels are not observable during training, we make an assumption on the prior distributions of test data. WLOG, we assume that the prior distribution of bounding box labels and geometric features are uniform. Note that we can adopt the KL-divergence for the implementation of $d_{\mathcal{A}}$ assuming that latent representations follow the univariate Gaussian distribution.*

**Connections with existing AL approaches.** The proposed criteria jointly optimize the discrepancy distance for both tasks with three objectives, which shows the connections with existing AL strategies. The uncertainty-based methods focus strongly on the first term, based on the assumption that learning more difficult samples will help to improve the suprema of the loss. This rigorous assumption can result in a bias towards hard samples, which will be accumulated and amplified across iterations. Diversity-based methods put more effort into minimizing the second term, aiming to align the distributions in the latent subspace. However, the diversity-based approaches are unable to discover the latent features specified for regression, which can be critical when dealing with a detection problem. We introduce the third term for the 3D detection task, motivated by the fact that aligning the geometric characteristics of point clouds helps to preserve the fine-grained details of objects, leading to more accurate regression. Our empirical study provided in Sec. 3.3 suggests jointly optimizing three terms can lead to the best performance.

## 2.3 OUR APPROACH

To optimize the three criteria outlined in Eq. 1, we derive an AL scheme consisting of three components. In particular, to reduce the computational overhead, we hierarchically filter the samples that meet the selection criteria (illustrated in Fig. 1): we first pick $\mathcal{K}_1$ candidates by concise label sampling (**Stage 1**), from which we select $\mathcal{K}_2$ representative prototypes (**Stage 2**), with $\mathcal{K}_1, \mathcal{K}_2 << n$. Finally, we leverage greedy search (**Stage 3**) to find the $N_r$ prototypes that match with the prior marginal distribution of test data. The hierarchical sampling scheme can save $\mathcal{O}((n - \mathcal{K}_1)T_2 + (n - \mathcal{K}_2)T_3)$ cost, with $T_2$ and $T_3$ indicating the runtime of criterion evaluation. The algorithm is summarized in the supplemental material. In the following, we describe the details of the three stages.

**Stage 1: Concise Label Sampling (CLS).** By using *label conciseness* as a sampling criterion, we aim to alleviate label redundancy and align the source label distribution with the target prior label distribution. Particularly, we find a subset $\mathcal{D}_{S_1}^*$ of size $\mathcal{K}_1$ that minimizes Kullback-Leibler (KL) divergence between the probability distribution $P_{Y_S}$ and the uniform distribution $P_{Y_T}$. To this end, we formulate the KL-divergence with Shannon entropy $H(\cdot)$ and define an optimization problem of maximizing the entropy of the label distributions:

$$D_{KL}(P_{\widehat{Y}_{S_1}} \| P_{Y_T}) = -H(\widehat{Y}_{S_1}) + \log |\widehat{Y}_{S_1}|, \tag{2}$$

$$\mathcal{D}_{S_1}^* = \underset{\mathcal{D}_{S_1} \subset \mathcal{D}_U}{\arg\min} D_{KL}(P_{\widehat{Y}_{S_1}} \| P_{Y_T}) = \underset{\mathcal{D}_{S_1} \subset \mathcal{D}_U}{\arg\max} H(\widehat{Y}_{S_1}), \tag{3}$$

where $\log |\widehat{Y}_{S_1}| = \log \mathcal{K}_1$ indicates the number of values $Y_{S_1}$ can take on, which is a constant. Note that $P_{Y_T}$ is a uniform distribution, and we removed the constant values from the formulations. We pass all point clouds $\{(\mathcal{P})_j\}_{i \in [n]}$ from the unlabeled pool to the detector and extract the predictive labels $\{\hat{y}_i\}_{i=1}^{N_B}$ for $N_B$ bounding boxes, with $\hat{y}_i = \arg\max_{y \in [C]} f(x_i; \boldsymbol{w}_f)$. The label entropy of the $j$-th point cloud $H(\widehat{Y}_{j,S})$ can be calculated as,

$$H(\widehat{Y}_{j,S}) = -\sum_{c=1}^{C} \boldsymbol{p}_{i,c} \log \boldsymbol{p}_{i,c}, \quad \boldsymbol{p}_{i,c} = \frac{e^{|\hat{y}_i = c|/N_B}}{\sum_{c=1}^{C} e^{|\hat{y}_i = c|/N_B}}. \tag{4}$$

Based on the calculated entropy scores, we filter out the top-$\mathcal{K}_1$ candidates and validate them through the **Stage 2** representative prototype selection.

**Stage 2: Representative Prototype Selection (RPS).** In this stage, we aim to to identify whether the subsets cover the *unique* knowledge encoded only in $\mathcal{D}_U$ and not in $\mathcal{D}_L$ by measuring the *feature representativeness* with gradient vectors of point clouds. Motivated by this, we find the representative prototypes on the gradient space $\mathcal{G}$ to form the subset $\mathcal{D}_{S_2}$, where magnitude and orientation represent the uncertainty and diversity of the new knowledge. For a classification problem, gradients can be retrieved by feeding the hypothetical label $\hat{y} = \arg\max_{y \in [C]} \boldsymbol{p}(y|x)$ to the networks. However, the gradient extraction for regression problem is not explored yet in the literature, due to the fact that the hypothetical labels for regression heads cannot be directly obtained. To mitigate this, we propose to enable Monte Carlo dropout (MC-DROPOUT) at the **Stage 1**, and get the averaging predictions $\bar{B}$ of $M$ stochastic forward passes through the model as the hypothetical labels for regression loss:

$$\bar{B} \approx \frac{1}{M} \sum_{i=1}^{M} g(\boldsymbol{x}; \boldsymbol{w}_d, \boldsymbol{w}_g), \boldsymbol{w}_d \sim \texttt{Bernoulli}(1-p), \tag{5}$$

$$G_{S_2} = \{\nabla_\Theta \ell^{reg}(g(\boldsymbol{x}), \bar{B}; \boldsymbol{w}_g), \boldsymbol{x} \sim \mathcal{D}_{S_2}\}, \tag{6}$$

with $p$ indicating the dropout rate, $\boldsymbol{w}_d$ the random variable of the dropout layer, and $\Theta$ the parameters of the convolutional layer of the shared block. The gradient maps $G_{S_2} \in \mathcal{G}$ can be extracted from shared layers and calculated by the chain rule. Since the gradients for test samples are not observable, we make an assumption that its prior distribution follows a Gaussian distribution, which allows us to rewrite the optimization function as,

$$\begin{aligned}
\mathcal{D}_{S_2}^* &= \arg\min_{\mathcal{D}_{S_2} \subset \mathcal{D}_{S_1}} D_{KL}(P_{X_{S_2}} \parallel P_{X_T}) \approx \arg\min_{\mathcal{D}_{S_2} \subset \mathcal{D}_{S_1}} D_{KL}(P_{G_{S_2}} \parallel P_{G_T}) \\
&= \arg\min_{\mathcal{D}_{S_2} \subset \mathcal{D}_{S_1}} \log \frac{\sigma_T}{\sigma_{S_2}} + \frac{\sigma_{S_2}^2 + (\mu_{S_2} - \mu_T)}{2\delta_T^2} - \frac{1}{2} \approx \mathcal{K}_2\text{-}\texttt{medoids}(G_{S_1}),
\end{aligned} \tag{7}$$

with $\mu_{S_2}$, $\sigma_{S_2}$ ($\mu_T$, and $\sigma_T$) being the mean and a standard deviation of the univariate Gaussian distribution of the selected set (test set), respectively. Based on Eq. 7, the task of finding a representative set can be viewed as picking $\mathcal{K}_2$ prototypes (*i.e.,* $\mathcal{K}_2$-medoids) from the clustered data, so that the centroids (mean value) of the selected subset and the test set can be naturally matched. The variance $\sigma_{S_2}$ and $\sigma_T$, basically, the distance of each point to its prototypes will be minimized simultaneously. We test different approaches for selecting prototypes in Sec. 3.3.

**Stage 3: Greedy Point Density Balancing (GPDB).** The third criterion adopted is *geometric balance*, which targets at aligning the distribution of selected prototypes with the marginal distribution of testing point clouds. As point clouds typically consist of thousands (if not millions) of points, it is computationally expensive to directly align the meta features (*e.g.,* coordinates) of points. Furthermore, in representation learning for point clouds, the common practice of using voxel-based architecture typically relies on quantized representations of point clouds and loses the object details due to the limited perception range of voxels. Therefore, we utilize the point density $\phi(\cdot, \cdot)$ within each bounding box to preserve the geometric characteristics of an object in 3D point clouds. By aligning the geometric characteristic of the selected set and unlabeled pool, the fine-tuned detector is expected to predict more accurate localization and size of bounding boxes and recognize both close (*i.e.,* dense) and distant (*i.e.,* sparse) objects at the test time. The probability density function (PDF) of the point density is not given and has to be estimated from the bounding box predictions. To this end, we adopt Kernel Density Estimation (KDE) using a finite set of samples from each class which can be computed as:

$$\boldsymbol{p}(\phi(\mathcal{P}, \widehat{\mathcal{B}})) = \frac{1}{N_B h} \sum_{j=1}^{N_B} \mathcal{K}er\left(\frac{\phi(\mathcal{P}, \widehat{\mathcal{B}}) - \phi(\mathcal{P}, \widehat{\mathcal{B}}_j)}{h}\right), \tag{8}$$

with $h > 0$ being the pre-defined bandwidth that can determine the smoothing of the resulting density function. We use Gaussian kernel for the kernel function $\mathcal{K}er(\cdot)$. With the PDF defined, the optimization problem of selecting the final candidate sets $\mathcal{D}_S$ of size $N_r$ for the label query is:

$$\mathcal{D}_S^* = \underset{\mathcal{D}_S \subset \mathcal{D}_{S_2}}{\arg\min} D_{KL}(\phi(\mathcal{P}_S, \widehat{\mathcal{B}}_S) \parallel \phi(\mathcal{P}_T, \mathcal{B}_T)), \tag{9}$$

where $\phi(\cdot, \cdot)$ measures the point density for each bounding box. We use greedy search to find the optimal combinations from the subset $\mathcal{D}_{S_2}$ that can minimize the KL distance to the uniform distribution $\boldsymbol{p}(\phi(\mathcal{P}_T, \mathcal{B}_T)) \sim \texttt{uniform}(\alpha_{lo}, \alpha_{hi})$. The upper bound $\alpha_{hi}$ and lower bound $\alpha_{lo}$ of the uniform distribution are set to the 95% density interval, *i.e.,* $\boldsymbol{p}(\alpha_{lo} < \phi(\mathcal{P}, \widehat{\mathcal{B}}_j) < \alpha_{hi}) = 95\%$ for every predicted bounding box $j$. Notably, the density of each bounding box is recorded during the **Stage 1**, which will not cause any computation overhead. The analysis of time complexity against other active learning methods is presented in Sec. 3.4.

# 3 EXPERIMENTS

## 3.1 EXPERIMENTAL SETUP

**Datasets.** KITTI (Geiger et al., 2012) is one of the most representative datasets for point cloud based object detection. The dataset consists of 3,712 training samples (*i.e.,* point clouds) and 3,769 *val* samples. The dataset includes a total of 80,256 labeled objects with three commonly used classes for autonomous driving: cars, pedestrians, and cyclists. The Waymo Open dataset (Sun et al., 2020) is a challenging testbed for autonomous driving, containing 158,361 training samples and 40,077 testing samples. The sampling intervals for KITTI and Waymo are set to 1 and 10, respectively.

**Generic AL Baselines**. We implemented the following five generic AL baselines of which the implementation details can be found in the supplementary material. (1) **RAND**: is a basic sampling method that selects $N_r$ samples at random for each selection round; (2) **ENTROPY** (Wang & Shang, 2014): is an *uncertainty*-based active learning approach that targets the *classification* head of the detector, and selects the top $N_r$ ranked samples based on the entropy of the sample's predicted label; (3) **LLAL** (Yoo & Kweon, 2019): is an *uncertainty*-based method that adopts an auxiliary network to predict an indicative loss and enables to select samples for which the model is likely to produce wrong predictions; (4) **CORESET** (Sener & Savarese, 2018): is a *diversity*-based method performing the core-set selection that uses the greedy furthest-first search on both labeled and unlabeled embeddings at each round; and (5) **BADGE** (Ash et al., 2020): is a *hybrid* approach that samples instances that are disparate and of high magnitude when presented in a hallucinated gradient space.

**Applied AL Baselines for 2D and 3D Detection**. For a fair comparison, we also compared three variants of the deep active learning method for 3D detection and adapted one 2D active detection method to our 3D detector. (6) **MC-MI** (Feng et al., 2019) utilized Monte Carlo dropout associated with mutual information to determine the uncertainty of point clouds. (7) **MC-REG**: Additionally, to verify the importance of the uncertainty in regression, we design an *uncertainty*-based baseline that determines the *regression* uncertainty via conducting $M$-round MC-DROPOUT stochastic passes at the test time. The variances of predictive results are then calculated, and the samples with the top-$N_r$ greatest variance will be selected for label acquisition. We further adapted two applied AL methods for 2D detection to a 3D detection setting, where (8) **LT/C** (Kao et al., 2018) measures the class-specific localization tightness, *i.e.*, the changes from the intermediate proposal to the final bounding box and (9) **CONSENSUS** (Schmidt et al., 2020) calculates the variation ratio of minimum IoU value for each RoI-match of 3D boxes.

## 3.2 COMPARISONS AGAINST ACTIVE LEARNING METHODS

**Quantitative Analysis**. We conducted comprehensive experiments on the KITTI and Waymo datasets to demonstrate the effectiveness of the proposed approach. The $\mathcal{K}_1$ and $\mathcal{K}_2$ are empirically set to 300 and 200 for KITTI and 2,000 and 1,200 for Waymo. Under a fixed budget of point clouds, the performance of 3D and BEV detection achieved by different AL policies are reported in Figure 2, with standard deviation of three trials shown in shaded regions. We can clearly observe that CRB consistently outperforms all state-of-the-art AL methods by a noticeable margin, irrespective of the number of annotated bounding boxes and difficulty settings. It is worth noting that, on the

Table 1: Performance comparisons (3D AP scores) with generic AL and applied AL for detection on KITTI *val* set with 1% queried bounding boxes.

| | Method | CAR | | | PEDESTRIAN | | | CYCLIST | | | AVERAGE | | |
|---|---|---|---|---|---|---|---|---|---|---|---|---|---|
| | | EASY | MOD. | HARD | EASY | MOD. | HARD | EASY | MOD. | HARD | EASY | MOD. | HARD |
| Generic | CORESET | 87.77 | 77.73 | 72.95 | 47.27 | 41.97 | 38.19 | 81.73 | 59.72 | 55.64 | 72.26 | 59.81 | 55.59 |
| | BADGE | 89.96 | 75.78 | 70.54 | 51.94 | 46.24 | 40.98 | 84.11 | 62.29 | 58.12 | 75.34 | 61.44 | 56.55 |
| | LLAL | 89.95 | 78.65 | **75.32** | 56.34 | 49.87 | 45.97 | 75.55 | 60.35 | 55.36 | 73.94 | 62.95 | 58.88 |
| AL-Det | MC-REG | 88.85 | 76.21 | 73.47 | 35.82 | 31.81 | 29.79 | 73.98 | 55.23 | 51.85 | 66.21 | 54.41 | 51.70 |
| | MC-MI | 86.28 | 75.58 | 71.56 | 41.05 | 37.50 | 33.83 | 86.26 | 60.22 | 56.04 | 71.19 | 57.77 | 53.81 |
| | CONSENSUS | 90.14 | 78.01 | 74.28 | 56.43 | 49.50 | 44.80 | 78.46 | 55.77 | 53.73 | 75.01 | 61.09 | 57.60 |
| | LT/C | 88.73 | 78.12 | 73.87 | 55.17 | 48.37 | 43.63 | 83.72 | 63.21 | 59.16 | 75.88 | 63.23 | 58.89 |
| | CRB | **90.98** | **79.02** | 74.04 | **64.17** | **54.80** | **50.82** | **86.96** | **67.45** | **63.56** | **80.70** | **67.81** | **62.81** |

Figure 2: 3D and BEV mAP (%) of CRB and AL baselines on the KITTI and Waymo *val* split.

KITTI dataset, the annotation time for the proposed CRB is 3 times faster than RAND, while achieving a comparable performance. Moreover, AL baselines for regression and classification tasks (*e.g.*, LLAL) or for regression only tasks (*e.g.*, MC-REG) generally obtain higher scores yet leading to higher labeling costs than the classification-oriented methods (*e.g.*, ENTROPY).

Table 1 reports the major experimental results of the state-of-the-art generic AL methods and applied AL approaches for 2D and 3D detection on the KITTI dataset. It is observed that LLAL and LT/C achieve competitive results, as the acquisition criteria adopted jointly consider the classification and regression task. Our proposed CRB improves the 3D mAP scores by 6.7% which validates the effectiveness of minimizing the generalization risk.

**Qualitative Analysis**. To intuitively understand the merits of our proposed active 3D detection strategy, Figure 3 demonstrates that the 3D detection results yielded by **RAND** (bottom left) and **CRB** selection (bottom right) from the corresponding image (upper row). Both 3D detectors are trained under the budget of 1K annotated bounding boxes. False positives and corrected predictions are indicated with red and green boxes. It is observed that, under the same condition, CRB produces more accurate and more confident predictions than RAND. Besides, looking at the cyclist highlighted in the orange box in Figure 3, the detector trained with RAND produces a significantly lower confidence score compared to our approach. This confirms that the samples selected by CRB are aligned better with the test cases. More visualizations can be found in the supplemental material.

## 3.3 ABLATION STUDY

**Study of Active Selection Criteria**. Table 2 reports the performance comparisons of six variants of the proposed CRB method and the basic random selection baseline (1-st row) on the KITTI dataset. We report the 3D and BEV mAP metrics at all difficulty levels with 1,000 bounding boxes annotated. We observe that only applying GPDB (4-th row) produces 12.5% lower scores and greater variance than the full model (the last row). However, with CLS (6-th row), the performance increases by approximately 10% with the minimum variance. This phenomenon evidences the importance of optimizing the discrepancy for both classification and regression tasks. It's further shown that re-

Table 2: Ablative study of different active learning criteria on the KITTI *val* split. 3D and BEV AP scores (%) are reported when 1,000 bounding boxes are annotated.

| CLS | RPS | GPDB | \multicolumn{3}{c}{3D Detection mAP} | | | \multicolumn{3}{c}{BEV Detection mAP} | | |
| | | | EASY | MODERATE | HARD | EASY | MODERATE | HARD |
| --- | --- | --- | --- | --- | --- | --- | --- | --- |
| - | - | - | $70.70_{\pm1.60}$ | $58.27_{\pm1.04}$ | $54.69_{\pm1.30}$ | $75.37_{\pm1.65}$ | $64.54_{\pm1.69}$ | $61.36_{\pm1.61}$ |
| ✓ | - | - | $77.76_{\pm1.70}$ | $64.56_{\pm1.39}$ | $59.54_{\pm1.13}$ | $81.07_{\pm1.67}$ | $69.76_{\pm1.45}$ | $65.01_{\pm1.31}$ |
| - | ✓ | - | $74.93_{\pm3.11}$ | $61.65_{\pm1.95}$ | $57.70_{\pm1.52}$ | $78.85_{\pm2.31}$ | $67.07_{\pm1.36}$ | $63.47_{\pm1.21}$ |
| - | - | ✓ | $69.11_{\pm13.22}$ | $56.12_{\pm12.74}$ | $52.85_{\pm11.49}$ | $73.57_{\pm10.45}$ | $62.49_{\pm10.62}$ | $59.45_{\pm9.78}$ |
| ✓ | ✓ | - | $76.19_{\pm2.13}$ | $62.81_{\pm1.31}$ | $58.03_{\pm1.18}$ | $80.73_{\pm0.92}$ | $68.67_{\pm0.21}$ | $64.42_{\pm0.22}$ |
| ✓ | - | ✓ | $76.72_{\pm0.78}$ | $64.70_{\pm1.07}$ | $59.68_{\pm0.93}$ | $80.71_{\pm0.26}$ | $70.01_{\pm0.40}$ | $65.47_{\pm0.56}$ |
| ✓ | ✓ | ✓ | $\mathbf{79.03}_{\pm1.39}$ | $\mathbf{65.86}_{\pm1.21}$ | $\mathbf{61.06}_{\pm1.43}$ | $\mathbf{82.60}_{\pm1.34}$ | $\mathbf{70.74}_{\pm0.57}$ | $\mathbf{66.41}_{\pm1.22}$ |

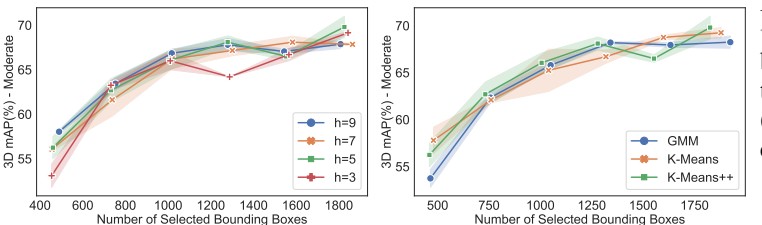

Figure 3: A case study of active 3D detection performance of **RAND** (bottom left) and **CRB** (bottom right) under the budge of 1,000 annotated bounding boxes. False positive (corrected predictions) are highlighted in red (green) boxes. The orange box denotes the detection with low confidence.

moving any selection criteria from the proposed CRB triggers a drop on mAP scores, confirming the importance of each in a sample-efficient AL strategy.

**Sensitivity to Prototype Selection.** We examine the sensitivity of performance to different prototype selection methods used in the RPS module on the KITTI dataset (moderate difficulty level). Particularly, In Figure 4 (right), we show the performance of our approach using different prototype selection methods of the Gaussian mixture model (GMM), K-MEANS, and K-MEANS++. To fairly reflect the trend in the performance curves, we run two trials for each prototype selection approach and plot the mean and the variance bars. K-MEANS is slightly more stable than the other two, with higher time complexity and better representation learning. It is observed that there is very little variation ($\sim 1.5\%$) in the performance of our approach when using different prototype selection methods. This confirms that the CRB's superiority over existing baselines is not coming from the prototype selection method.

Figure 4: Results on KITTI *val* set with varying KDE bandwidth $h$ (left) and prototype selection approaches (right) with increasing queried bounding boxes.

**Sensitivity to Bandwidth $h$.** Figure 4 depicts the results of CRB with the bandwidth $h$ varying in $\{3, 5, 7, 9\}$. Choosing the optimal bandwidth value $h^*$ can avoid under-smoothing ($h < h^*$) and over-smoothing ($h > h^*$) in KDE. Except $h = 3$ which yields a large variation, CRB with the bandwidth of all other values reach similar detection results within the 2% absolute difference on 3D mAP. This evidences that the CRB is robust to different values of bandwidth.

**Sensitivity to Detector Architecture**. We validate the sensitivity of performance to choices of one-stage and two-stage detectors. Table 4 reports the results with the SECOND detection backbone on the KITTI dataset. With only 3% queried 3D bounding boxes, it is observed that the proposed CRB approach consistently outperforms the SOTA generic active learning approaches across a range of detection difficulties, improving 4.7% and 2.8% on 3D mAP and BEV mAP scores.

Table 3: Performance comparisons on KITTI *val* set *w.r.t.* varying thresholds $\mathcal{K}_1$ and $\mathcal{K}_2$ after two rounds of active selection (8% point clouds, 1% bounding boxes). Results are reported with 3D AP with 40 recall positions. $^{\dagger}$ indicates the reported performance of the backbone trained with the full labeled set (100%).

| $\mathcal{K}_1$ | $\mathcal{K}_2$ | CAR | | | PEDESTRIAN | | | CYCLIST | | | AVERAGE | | |
|---|---|---|---|---|---|---|---|---|---|---|---|---|---|
| | | EASY | MOD. | HARD | EASY | MOD. | HARD | EASY | MOD. | HARD | EASY | MOD. | HARD |
| 500 | 400 | 90.04 | 79.08 | **74.66** | 57.11 | 51.10 | **51.12** | 81.97 | 63.40 | 59.62 | 76.50 | 64.53 | 60.10 |
| 500 | 300 | 90.98 | 79.02 | 74.04 | 64.17 | 54.80 | 50.82 | **86.96** | **67.45** | **63.56** | **80.70** | **67.81** | **62.81** |
| 400 | 300 | **91.30** | **79.21** | 74.00 | 62.93 | 55.67 | 49.27 | 79.02 | 60.50 | 56.74 | 77.75 | 65.13 | 60.00 |
| 300 | 200 | 90.45 | 78.81 | 73.44 | **65.00** | **55.91** | **51.12** | 84.82 | 65.77 | 61.53 | 80.09 | 67.32 | 62.05 |
| PV-RCNN$^{\dagger}$ | | 92.56 | 84.36 | 82.48 | 64.26 | 56.67 | 51.91 | 88.88 | 71.95 | 66.78 | 81.75 | 70.99 | 67.06 |

**Sensitivity Analysis of Thresholds $\mathcal{K}_1$ and $\mathcal{K}_2$.** We examine the sensitivity of our approach to different values of threshold parameters $\mathcal{K}_1$ and $\mathcal{K}_2$. We report the mean average precision (mAP) on the KITTI dataset, including both 3D and BEV views at all difficulty levels. We check four possible combinations of $\mathcal{K}_1$ and $\mathcal{K}_2$ and show the results in Table 3. We can observe that at MODERATE and HARD levels, there is only 3.28% and 2.81% fluctuation on average mAP. In the last row, we further report the accuracy achieved by the backbone detector trained with all labeled training data and a larger batch size. With only 8% point clouds and 1% annotated bounding boxes, CRB achieves a comparable performance to the full model.

## 3.4 COMPLEXITY ANALYSIS

Table 5 shows the time complexity of training and active selection for different active learning approaches. $n$ indicates the total number of unlabeled point clouds, $N_r$ is the quantity selected, and $E$ is the training epochs, with $N_r \ll n$. We can clearly observe that, at training stage, the complexity of all AL strategies is $\mathcal{O}(En)$, except LLAL that needs extra epochs $E_l$ to train the loss prediction module. At the active selection stage, RAND randomly generates $N_r$ indices to retrieve samples from the pool. CORESET computes pairwise distances between the embedding of selected samples and unlabeled samples that yields the time complexity of $\mathcal{O}(N_r n)$. BADGE iterates through the gradients of all unlabeled samples passing gradients into K-MEANS++ algorithm, with the complexity of $\mathcal{O}(N_r n)$ bounded by K-MEANS++. Given $\mathcal{K}_1, \mathcal{K}_2 \approx N_r$, the time complexity of our method is $\mathcal{O}(n \log n + 2N_r^2)$, with $\mathcal{O}(n \log(n))$ being the complexity of sorting the entropy scores in CLS, and $\mathcal{O}(N_r^2)$ coming from $\mathcal{K}_2$-medoids and greedy search in RPS and GPDB. Note that, in our case, $\mathcal{O}(n \log n + 2N_r^2) < \mathcal{O}(N_r n)$. The complexity of simple ranking-based baselines is $\mathcal{O}(n \log(n))$ due to sorting the sample acquisition scores. Comparing our method with recent state-of-the-arts, LLAL has the highest training complexity, and BADGE and CORESET have the highest selection complexity. Unlike the existing baseline, training and selection complexities of the proposed CRB are upper bounded by the reasonable asymptotic growth rates.

Table 4: AL Results with one-stage 3D detector SECOND.

| | 3D Detection mAP | | | BEV Detection mAP | | |
|---|---|---|---|---|---|---|
| | EASY | MODERATE | HARD | EASY | MODERATE | HARD |
| RAND | 75.23 | 60.83 | 56.55 | 80.20 | 67.56 | 63.30 |
| LLAL | 72.02 | 58.96 | 54.21 | 79.50 | 66.82 | 62.48 |
| CORESET | 74.74 | 58.86 | 54.61 | 79.71 | 65.53 | 61.39 |
| BADGE | 75.38 | 61.65 | 56.72 | 80.81 | 68.83 | 64.17 |
| CRB | **78.96** | **64.27** | **59.60** | **83.28** | **70.49** | **66.09** |

Table 5: Complexity Analysis.

| AL Strategy | Training | Selection |
|---|---|---|
| RAND | $\mathcal{O}(En)$ | $\mathcal{O}(N_r)$ |
| ENTROPY | $\mathcal{O}(En)$ | $\mathcal{O}(n \log n)$ |
| MC-REG | $\mathcal{O}(En)$ | $\mathcal{O}(n \log n)$ |
| LLAL | $\mathcal{O}((E + E_l)n)$ | $\mathcal{O}(n \log n)$ |
| CORESET | $\mathcal{O}(En)$ | $\mathcal{O}(N_r n)$ |
| BADGE | $\mathcal{O}(En)$ | $\mathcal{O}(N_r n)$ |
| CRB | $\mathcal{O}(En)$ | $\mathcal{O}(n \log n + 2N_r^2)$ |

## 4 DISCUSSION

This paper studies three novel criteria for sample-efficient active 3D object detection, that can effectively achieve high performance with minimum costs of 3D box annotations and runtime complexity. We theoretically analyze the relationship between finding the optimal acquired subset and mitigating the sets discrepancy. The framework is versatile and can accommodate existing AL strategies to provide in-depth insights into heuristic design. The limitation of this work lies in a set of assumptions made on the prior distribution of the test data, which could be violated in practice. For more discussions, please refer to Sec. A.1 in Appendix. In contrast, it opens an opportunity of adopting our framework for active domain adaptation, where the target distribution is accessible for alignment. Addressing these two avenues is left for future work.

ACKNOWLEDGEMENT

This work was supported by Australian Research Council (CE200100025).

REPRODUCIBILITY STATEMENT

The source code of the developed active 3D detection toolbox is available in the supplementary material, which accommodates various AL approaches and one-stage and two-stage 3D detectors. We specify the settings of hyper-parameters, the training scheme and the implementation details of our model and AL baselines in Sec. B of the supplementary material. We show the proofs of Theorem 2.1 in Sec. C followed by the overview of the algorithm in Sec. D in the supplementary material. We repeat the experiments on the KITTI dataset 3 times with different initial labeled sets and show the standard deviation in plots and tables.

ETHICS STATEMENT

Our work may have a positive impact on communities to reduce the costs of annotation, computation, and carbon footprint. The high-performing AL strategy greatly enhances the feasibility and practicability of 3D detection in critical yet data-scarce fields such as medical imaging. We did not use crowdsourcing and did not conduct research with human subjects in our experiments. We cited the creators when using existing assets (*e.g.*, code, data, models).

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
