# OpenReview forum: "Exploring Active 3D Object Detection from a Generalization Perspective"
_ICLR.cc/2023/Conference — ICLR 2023 notable top 25%_

### Official Review · Reviewer_BMpC · 2022-10-20

**Confidence:** 4
**Correctness:** 3
**Technical Novelty And Significance:** 3
**Empirical Novelty And Significance:** 3
**Recommendation:** 8

**Clarity, Quality, Novelty And Reproducibility:**

Heading 3.2 (line 3): I would suggest to use "300 and 200 for KITTI" instead of "300, 200 for KITTI" since the latter is easily mistaken for 300200.

**Strength And Weaknesses:**

Strengths:

The paper is well written, the experiments are convincing, and the proposed method is simple to implement. Figure 1 nicely illustrates the overall framework of the method.

The work presents an exhaustive experimental evaluation which demonstrates the effectiveness of the presented approach on KITTI and Waymo datasets.

Ablation study (Table 2) and complexity analysis  (Table 4) support the experimental well.

Weaknesses:

Although the complexity analysis is given in the article, no specific time cost is given.

These experiments are all carried out under the autonomous driving dataset. Does this structure work for the 3D detection task of the indoor dataset?

**Summary Of The Paper:**

This paper studies the problem of active 3D object detection from point clouds. The main problem addressed in this paper is how to design a proper criterion for cost-efficient sample acquisition. In this work, the authors jointly investigate three novel criteria in the framework CRB for point cloud acquisition (label conciseness, feature representativeness and geometric balance) which hierarchically filters out the point clouds of redundant 3D bounding box labels.

In particular, (1)To alleviate the issues of label redundancy and class imbalance, the entropy of bounding box label predictions are used to pick top *K*1 point clouds. (2)Monte Carlo dropout and the hypothetical labels by averaging predictions from multiple stochastic forward passes are used for considering the impact of classification and regression objectives on gradients. (3) Finally, to maintain the *geometric balance* property, we minimize the KL divergence between the marginal distributions of point cloud density of each predicted bounding box.


**Summary Of The Review:**

This paper is overall a goog paper, my main concern is the above weakness .
The author's response addressed my concerns well, so my final recommendation is accept

---

> ### Author Response · Authors · 2022-11-09
> **Initial Response to Reviewer BMpC**
>
> We thank the reviewer for the positive and constructive comments. Please find our response to each comment below:
>
> > W1 Although the complexity analysis is given in the article, no specific time cost is given.
>
>
> **Re W1**. Thanks for the constructive comment. To answer your question, we have checked the experiments or running all AL baselines and the derived CRB approach with the SECOND backbone on the KITTI dataset. In the following table, we report the total time costs of 6 rounds of active selection and model training for 240 epochs in total.  Please note that **the general trend of the reported time costs of baselines and our method aligns with our complexity analysis**, except CORESET runs faster than expected and approximates the speed of random sampling.  We can observe that the proposed CRB is faster than BADGE and achieve a similar time cost as LLAL.
>
> | Method    | RAND     | CORESET   | LLAL      | CRB       | BADGE      |
> |-----------|------------|-----------|-----------|-----------|------------|
> | Time Cost | 4h 17m 18s | 4h 19m 2s | 4h 57m 7s | 5h 2m 19s | 5h 22m 33s |
>
> We summarized the following **3 reasons** that cause a slight inconsistency between the time costs and complexity analysis,
> 1) From our implementation code, CORESET computes the feature distances between embeddings on GPUs directly, while BADGE and our CRB method both detach the gradients to CPUs and the memory first for computation. We reckon this is the main reason that results in CORESET being significantly faster.
> 2) The total running time for each method includes other activities such as wandb connections, and log print out.
> 3) Some baseline methods are implemented based on different libraries (e.g., scikit-learn, scipy), which adopt various acceleration strategies and cause computational differences even for the same algorithm. For example, most scikit-learn models are implemented either with compiled Cython extensions or optimized computing libraries. However, our greedy search algorithm is purely implemented by ourselves and does not rely on any existing libraries, which is relatively time-consuming. Therefore, there is room available to further optimize the implementation of our method for more acceleration gains in future work.
>
>
>
> > W2 These experiments are all carried out under the autonomous driving dataset. Does this structure work for the 3D detection task of the indoor dataset?
>
> **Re W2.** Thanks for raising the important question. Theoretically, the proposed method **is capable of being effective on both outdoor and indoor point cloud datasets**. Recall that our proposed approach consists of three modules: Concise Label Sampling (CLS), Representative Prototype Selection (RPS) and Greedy Point Density Balancing (GPDB). CLS aims to alleviate the issues of label redundancy and class imbalance, which are also common in indoor datasets. For example, in ScanNet [1] datasets, 27.7% of the objects belong to “chair”, while some rare categories such as “bathtub”, “sink”, “bed”, etc. account for less than 2% [2]. CLS can address this issue effectively by reducing the redundancy of “chair” and acquiring more samples of rare categories. RPS is a task-agnostic module that works for both outdoor and indoor scenarios by exploring the feature representativeness on the gradient space. GPDB maintains the geometric balance of selected objects. Even though the indoor point cloud is relatively more evenly distributed compared to the outdoor scenes, the geometric imbalance issue caused by object occlusion is still commonly observed, which makes the trained 3D detector hard to generalize [3]. GPDB is a suitable solution to address such an issue. Therefore, the proposed three modules of CRB are still valid and reasonable for indoor datasets.
>
> To support our claims, we are happy to adapt all AL baselines and the proposed CRB approach to indoor datasets. However, due to the time constraints and the limitation that the current detector backbones such as SECOND and PV-RCNN do not have specific configurations (*e.g.*, anchor sizes) for indoor datasets, we plan to improve our active 3D detection toolbox and enable the support of indoor datasets by referring to MMDetection3D [4]. After upgrading the toolbox, we will then conduct the experiments as part of future work, and provide results in the revised version.
>
>
> > W3. Heading 3.2 (line 3): I would suggest to use "300 and 200 for KITTI" instead of "300, 200 for KITTI" since the latter is easily mistaken for 300200.
>
> **Re W3.** Thanks for pointing out the writing with ambiguity. We have fixed it in the revised version.

---

> > ### Author Response · Authors · 2022-11-09
> > **Initial Response to Reviewer BMpC - References**
> >
> > Due to the limit of characters, we include the reference list of our response below:
> > ### References
> > [1] Dai, A., Chang, A. X., Savva, M., Halber, M., Funkhouser, T., & Nießner, M. (2017). Scannet: Richly-annotated 3d reconstructions of indoor scenes. In Proceedings of the IEEE conference on computer vision and pattern recognition (pp. 5828-5839).
> >
> > [2] Cendra, F. J., Ma, L., Shen, J., & Qi, X. (2022). SL3D: Self-supervised-Self-labeled 3D Recognition. arXiv preprint arXiv:2210.16810.
> >
> > [3] Liang, J., An, P., & Ma, J. (2022). Distribution Aware VoteNet for 3D Object Detection. in AAAI 2022.
> >
> > [4] Contributors, MMDetection3D. "MMDetection3D: OpenMMLab next-generation platform for general 3D object detection."(2020). https://github.com/open-mmlab/mmdetection3d

---

### Official Review · Reviewer_S6aF · 2022-10-23

**Confidence:** 3
**Correctness:** 3
**Technical Novelty And Significance:** 3
**Empirical Novelty And Significance:** 4
**Recommendation:** 8

**Clarity, Quality, Novelty And Reproducibility:**

This paper is well-written, and seems solid.

There are a number of literatures on active learning in 2D object detection. Referring to the development of general 2D detectors to 3D detectors, this paper seems adapt data based on 3D point cloud data and 3D bounding box labeling, and the novelty is not well-clarified.

**Strength And Weaknesses:**

### Srength:
- Activate learning in point cloud object detection is interesting and novel. I believe this paper can provide insights and inspiration to the related works in the 3D object detection area.

- The theoretical analysis is sufficient and helpful to understand the method, and also provide support for the final performance validation.

- The experiments are extensive and can validate the effectiveness of the proposed framework.


### Weaknesses:
- I would suggest the authors clearly clarify the major difference of the paper with existing works on object detection and point clouds processing. It seems that this paper is analyzed in an general optimized view, but it will be better if the author can explicitly illusrate the innovation and the fundamental difference with exisiting techniques.

- For related work, I would suggest that the author add literature on 2D Detection and some of the latest works in 3D point cloud detectors. Just name a few:

	1. QBox: Partial Transfer Learning With Active Querying for Object Detection [2021, TNNLS]
	2. Towards Dynamic and Scalable Active Learning with Neural Architecture Adaption for Object Detection [2021, BMVC]: Add NAS into the AL loops.
	3. Entropy-based Active Learning for Object Detection with Progressive Diversity Constraint [2022, CVPR]
	4. Weakly Supervised Object Detection Based on Active Learning [2022, NPL]
	5. Active Learning Strategies for Weakly-Supervised Object Detection [2022]
	6. Label-Efficient Point Cloud Semantic Segmentation: An Active Learning Approach [2021, CVPR]
	7. ReDAL: Region-based and Diversity-aware Active Learning for Point Cloud Semantic Segmentation [2021, ICCV]
	8. Active Learning for Point Cloud Semantic Segmentation via Spatial-Structural Diversity Reasoning [2022]

- As for the experimental results in Table 1, it is a little bit counterintuitive: the proposed CRB with 1% labels can perform better than the general full-label training model. In my view, the active learning methods can achieve the performance very close to the full-label training model. The authors are suggested to explicitly clarify why the performance is even better than fully supervised model.

-More experiments on various detectors (e.g., SECOND, PV-RCNN) are required to illustrate the effectiveness of the proposed criteria for general active 3D object detection. Additionally, indoor scenarios such as ScanNet should also be taken into consideration.

**Summary Of The Paper:**

This paper explores how activate learning can help reduce annotation costs in the field of object detection of 3D point clouds. Specifically, to achieve efficient active learning with limited fixed annotation budgets, three selection criteria termed CRB are proposed to learn better sample acquisition of the 3D boxes annotation. Extensive experiments on KITTI and Waymo validate the effectiveness of the proposed method. The paper is well-written and easy to follow. The experimental results demonstrate the effectiveness of the proposed hierarchical active learning scheme in active learning for 3D object detection.


**Summary Of The Review:**

I'm not quite familiar with this topic, but this paper looks very comprehensive and solid, so I'm inclined to borderline accept.

---

> ### Author Response · Authors · 2022-11-09
> **Initial Response to Reviewer S6aF**
>
> We thank the reviewer for the valuable comments and answer them, as numbered, below:
>
> > W1. I would suggest the authors clearly clarify the major difference of the paper with existing works on object detection and point clouds processing. It seems that this paper is analyzed in a general optimized view, but it will be better if the author can explicitly illustrate the innovation and the fundamental difference with exisiting techniques.
>
> Re W1. Thanks for the constructive suggestion. We would like to summarize the major contributions of this work below:
>
> From a high level, different from existing works, our work is the **first comprehensive study** on active 3D detection, in which rigorous baselines and active learning protocol are set up and implemented to ensure a fair comparison of detection performance against annotation budget **at a bounding box level**. It is non-trivial to consider the box-level annotations, since the previous AL methods generally favor the point clouds with more objects, which have a higher chance of containing uncertain and diverse objects. Instead of following the heuristics of measuring diversity and uncertainty, our work focuses on the ultimate goal of active 3D detection - **a more generalizable detector can perform well at the test time**. This motivates us to design three selection criteria to determine the most informative point clouds without having redundant, or previously seen, or geometrically similar objects for label acquisition.
>
>
> Below, we will discuss the major difference of the derived methods from existing active 3D detection and active 2D detection approaches. Please note that we have  included a comprehensive review of those two groups of methods in Section I of the supplementary material.
>
>
> ----
>
> Active learning for 2D and 3D object detection has been relatively under-explored than the ones for image classification. Most existing active learning approaches for detection utilized hand-crafted heuristics such as Shannon entropy and localization tightness that were designed for traditional image classification tasks.
>
> - **[CLS]** Technically, the concise label sampling in Stage 1 also leverages Shannon entropy while the motivation and implementations are **different** from entropy based AL methods. ENTROPY approaches calculate the entropy value of each bounding box’s prediction and then rank the averaged entropy scores for each point cloud. They aim to identify point clouds containing uncertain objects while implicitly favoring hard samples. However, our CLS module is motivated by matching the label distribution of the ideal test set, which can help filter out the point clouds containing semantically redundant information and save annotation costs. For each point cloud, we obtain the hypothetical box labels by applying the argmax operations on logits, and calculate the frequency distribution of hypothetical labels, which is then utilized to calculate the entropy scores. **Our CLS modules focus on aligning label distribution rather than measuring prediction uncertainty in previous works.**
>
> - **[RPS]** Object detection is a combined task of bounding box classification and regression. Despite the former part having been well studied, mining the point clouds that the model cannot well regress remains as an underexplored problem. Localization tightness based AL methods calculate overlapping areas between region proposals and the final predictions of bounding boxes as an indicator of regression uncertainty. While effective, this group of methods **can only be applied to two-stage detectors**, of which the feasibility is highly restricted. Our approach differs from prior works in a sense that it **explores the regression and classification quality in the gradient space**: we first construct the hypothetical labels by leveraging Monte Carlo dropout and get the average predictions (*e.g.*, the position and orientation of detected bounding boxes) through multiple stochastic forward passes as the hypothetical labels for the regression loss. Then we use the chain rule to obtain the latent gradient map and run K-medoids to select the representative prototypes as the candidates. **This ensures the selected bounding boxes are previously unseen and non-repeated, which will be highly likely to produce erroneous or high-variance regression at the test time**. To this end, querying these samples to get an extra supervision can potentially benefit the regression performance of test samples without triggering high annotation costs. Additionally, the derived RPS strategy can accommodate both one-stage and two-stage detectors.

---

> > ### Author Response · Authors · 2022-11-09
> > **Initial Response to Reviewer S6aF  [Cont']**
> >
> >
> > - **[GPDB]** In the third stage, our approach takes into account **the unique geometric property of 3d point clouds**, *i.e.*, point cloud density, and improves the generalization of the model from a novel perspective of distribution alignment. In order to allow the queried instances from the selected point clouds to be complementary to each other, we crop the original point clouds based on the predicted bounding boxes and leverage the kernel density estimation (KDE) to estimate the probability density function, making sure it aligns well with the prior distribution of the ideal test sets.
> >
> > Beyond the technical contributions in aligning label distribution, jointly considering both tasks and the unique geometric property of 3D point clouds, we additionally analyze the time costs of point cloud selection and training. The derived hierarchical filtering strategy not only seamlessly integrates all criterion into a joint framework but also **control the time complexity** to be lower than AL strategies like CORESET and BADGE.
> >
> >
> >
> > > W2. For related work, I would suggest that the author add literature on 2D Detection and some of the latest works in 3D point cloud detectors.
> >
> > **Re W2.** Thanks for the constructive suggestions, and we agree that the above-mentioned works are important and related to this work, although the main tasks are a bit different. We have further included the discussion of these related works in the revised supplementary material accordingly (Section I, Page 9-10) which are highlighted in blue.
> >
> > > W3. As for the experimental results in Table 1, it is a little bit counterintuitive: the proposed CRB with 1% labels can perform better than the general full-label training model. In my view, the active learning methods can achieve the performance very close to the full-label training model. The authors are suggested to explicitly clarify why the performance is even better than fully supervised model.
> >
> > **Re W3.** Thanks for your comment. We would like to clarify that we stated “CRB achieves a **comparable** performance to the full model” in Section 3.3 Sensitivity Analysis of Thresholds K1 and K2, and we did not overclaim it to be superior to the full model. We observed that there is a performance gap between the detector learned with 1% bounding boxes and the one with the full training set, *i.e.*, 1.05%, 3.18% and 4.25% at the difficulty level of EASY, MODERATE and HARD, respectively.
> >
> > > W4. More experiments on various detectors (e.g., SECOND, PV-RCNN) are required to illustrate the effectiveness of the proposed criteria for general active 3D object detection. Additionally, indoor scenarios such as ScanNet should also be taken into consideration.
> >
> >
> > **Re W4.** Thanks for raising the important question. Theoretically, the proposed method is capable of being effective on both outdoor and indoor point cloud datasets. Recall that our proposed approach consists of three modules: Concise Label Sampling (CLS), Representative Prototype Selection (RPS) and Greedy Point Density Balancing (GPDB). CLS aims to alleviate the issues of label redundancy and class imbalance, which are also common in indoor datasets. For example, in ScanNet [1] datasets, 27.7% of the objects belong to “chair”, while some rare categories such as “bathtub”, “sink”, “bed”, etc. account for less than 2% [2]. CLS can address this issue effectively by reducing the redundancy of “chair” and acquiring more samples of rare categories. RPS is a task-agnostic module that works for both outdoor and indoor scenarios by exploring the feature representativeness on the gradient space. GPDB maintains the geometric balance of selected objects. Even though the indoor point cloud is relatively more evenly distributed compared to the outdoor scenes, the geometric imbalance issue caused by object occlusion is still commonly observed, which makes the trained 3D detector hard to generalize [3]. GPDB is a suitable solution to address such an issue. Therefore, the proposed three modules of CRB are still valid and reasonable for indoor datasets.

---

> > > ### Author Response · Authors · 2022-11-09
> > > **Initial Response to Reviewer S6aF [Cont']**
> > >
> > >
> > > To support our claims, we are happy to adapt all AL baselines and the proposed CRB approach to indoor datasets. However, due to the time constraints and the limitation that the current detector backbones such as SECOND and PV-RCNN do not have specific configurations (*e.g.*, anchor sizes) for indoor datasets, we plan to improve our active 3D detection toolbox and enable the support of indoor datasets by referring to MMDetection3D [4]. After upgrading the toolbox, we will then conduct the experiments as part of future work, and provide results in the revised version.
> > >
> > > ### References
> > > [1] Dai, A., Chang, A. X., Savva, M., Halber, M., Funkhouser, T., & Nießner, M. (2017). Scannet: Richly-annotated 3d reconstructions of indoor scenes. In Proceedings of the IEEE conference on computer vision and pattern recognition (pp. 5828-5839).
> > >
> > > [2] Cendra, F. J., Ma, L., Shen, J., & Qi, X. (2022). SL3D: Self-supervised-Self-labeled 3D Recognition. arXiv preprint arXiv:2210.16810.
> > >
> > > [3] Liang, J., An, P., & Ma, J. (2022). Distribution Aware VoteNet for 3D Object Detection.
> > >
> > > [4] Contributors, MMDetection3D. "MMDetection3D: OpenMMLab next-generation platform for general 3D object detection."(2020). https://github.com/open-mmlab/mmdetection3d

---

### Official Review · Reviewer_8qfL · 2022-10-25

**Confidence:** 4
**Correctness:** 3
**Technical Novelty And Significance:** 3
**Empirical Novelty And Significance:** 3
**Recommendation:** 6

**Clarity, Quality, Novelty And Reproducibility:**

Questions:
1. Why does GPDB alone in Table 2 has so wide an error bar and underperform the baseline?
2. Why different lines in Figure 2 have different sample point w.r.t Number of Selected Bounding Boxes ?

**Strength And Weaknesses:**

Pros:
1. The authors gave a good summary of existing active learning methods for LiDAR object detection, which highlighted the significance and popularity of the problem under study.
2. The authors developed an active-3D-det toolbox which could be quite helpful for setting fair and rigorous baselines for future researches.
3. Experiments on KITTI showed promising results.

Cons:
1. There are quite some factual errors in the method part of this work.
    1. PointNet series did not encode free-point cloud into a 3D tensor. On the bottom of Page 2, “Mainstream 3D object detectors encode free-form point clouds P into a regular feature map x ∈ R W ·L·F directly (Qi et al., 2017) or by voxelization (Shi et al., 2020), with W, L, F representing width, length, and channels of the feature map.”
    2. Wrong assumptions about loss function. The authors assumed that “The loss functions for classification and regression are assumed to be Lipschitz continuous”, but a simple binary cross entropy loss is not Lipschitz continuous for p in (0, beta]. Also I failed to connect the Lipschitz continuous of loss function with anything in Section 2.2.
2. Section 2.2 is quite vague to me, I don’t how Theorm 2.1 is connected to Eq 1 and what are Ps in Eq 1
3. A uniform prior of target labels as assumed in Eq. 2 is too far from the observed reality in LiDAR object detection
4. Stage 2 in section 2.3 is quite hard to understand. Each step in Eq. 7 seems too big a leap for me and proper discussion are not made in the text part.
5. Only the KITTI experiments showed promising results. For the Waymo experiments in Figure 2, CRB is out-performed by LLAL and ENTROPY in the low data regime while almost on par with RAND baseline in the high data regime. I find there are implementations for other datasets like nuScenes and Lyft in the supplementary codes, which makes me wonder if the method generalizes well across different datasets.

**Summary Of The Paper:**

The authors proposed three new criteria for 3D box-level active learning in LiDAR object detection. The first part(Eq. 4) encourages selecting more class-balance point clouds. The second part advocates selecting point clouds with more diverse gradients w.r.t. pseudo classification and regression labels. The las thet part favors point clouds with similar object point density as of the labelled set. Empirical studies showed the proposed three components are mostly effective and out-perform existing active learning methods on some datasets.

**Summary Of The Review:**

It is easy to see that the authors put a lot of efforts in this work and tried to connect the proposed selection criteria with some theoretical supports. Although there are some errors and unclear writings in the method part, but the proposed methods are backed up by at least solid experiments on the KITTI datasets, so I am leaning to recommend for its acceptance.

---

> ### Author Response · Authors · 2022-11-09
> **Initial Response to Reviewer 8qfL**
>
>
> We thank the reviewer for the valuable comments and answer them, as numbered, below:
>
> > W1. PointNet series did not encode free-point cloud into a 3D tensor. On the bottom of Page 2, “Mainstream 3D object detectors encode free-form point clouds P into a regular feature map x ∈ R W ·L·F directly (Qi et al., 2017) or by voxelization (Shi et al., 2020), with W, L, F representing width, length, and channels of the feature map.”
>
> **Re W1.** Thanks for your comment. In a typical point-based structure such as PointRCNN, there is a learned feature map with a size of (256, 512, 3+1), before feeding into the classification and regression head. The first dimension indicates the number of ROI, the second indicates the number of sampled points in each ROI, while the last one indicates the xyz and intensity. But we do apologize if the reference for the point-based methods group was misleading. We have rectified the descriptions in the updated manuscript (please refer to Section 2.1).
>
>
> > W2. Wrong assumptions about loss function. The authors assumed that “The loss functions for classification and regression are assumed to be Lipschitz continuous”, but a simple binary cross entropy loss is not Lipschitz continuous for p in (0, beta]. Also I failed to connect the Lipschitz continuous of loss function with anything in Section 2.2.
>
>
> **Re W2.** Thanks for pointing this out. We do agree with you that the standard cross-entropy loss is not strongly convex and Lipschitz continuous. However, in our model, the softmax function is combined with the cross-entropy loss. In line to [3], we regard softmax as the last layer of the DNN, and we assume the output $x$ of the network lies in the probability simplex. If the output, $z$ from the second to last layer of the DNN, which is the input to softmax, lies in a compact set, *i.e.*, $|z_j | \leq C$ for all $i$ and some $C$ > 0, then $softmax(z)_j ≥ e^{−2C}$, and so the range of softmax lies in the set
> $A := \{y \in \mathbb{R}^D : y_i \geq e^{−2C}, y_1 + · · · + y_D = 1\}$,
> which is strictly interior to the probability simplex. **Restricted to $A$, the cross-entropy loss is strongly convex and Lipschitz continuous**. We have revised the assumptions we made and cite the related work for reference in Section 2.1 as highlighted in blue.
>
>
> Here the Lipschitz continuity is a constraint on the classification loss and regression loss, which is added to infer the upper bound of the generalization risk of 3D detector. As Section 2.2 does not involve the detail of backbone training, so Lipschitz continuity presented is not highly relevant to any equations in Section 2.2. The details of the training losses for training 3D backbone detectors can be found in PV-RCNN [4] (Section 3.3) and SECOND [5] (Section 3.2.1).
>
>
> > W3. Section 2.2 is quite vague to me, I don’t how Theorm 2.1 is connected to Eq 1 and what are Ps in Eq 1
>
> **Re W3.** Thanks for your valuable comment. In order to achieve a tighter upper bound of the generalization risk of active 3D detection on the test set, we minimize the right side of the inequality in Theorem 2.1. The inequality comprises four terms: the training error, the discrepancy between the selected set and the test set, the joint risk and a constant. Due to the zero training assumption, we can only optimize the second term, which is the set discrepancy. This motivates us to construct the best optimal subset $\mathcal{D}_S^*$ by minimizing the discrepancy distance of empirical distributions of the two sets, so that the upper bound of generalization risks can be minimized at the same time.
>
> To mathematically define the discrepancy between the selected set and the test set, we set up **Definition 1** which considers the discrepancy distance w.r.t classification f and regression g. Considering the inputs and outputs for classification and regression, we use $d_{\mathcal{A}}$ to measure the discrepancy at the level of bounding box label prediction, latent features, and the point cloud density within each predicted bounding box. These three aspects bring us to form conciseness, representativeness, and balance in Equation (1), which can jointly consider the data discrepancy for both tasks.
>
> In addition, as defined in 2.1, $\mathcal{P}$ denotes a free-form point cloud. In this case, $\mathcal{P}_S$ and $\mathcal{P}_T$ represent the point clouds in the selected set and the ones in the test set. $\phi(\mathcal{P}, \hat{B}_S)$ indicates the density function for estimating the point cloud density of the predicted bounding box $\hat{B}_S$. To ease the understanding, we have added up more explanations in the Remark section.

---

> > ### Author Response · Authors · 2022-11-09
> > **Initial Response to Reviewer 8qfL [Cont']**
> >
> >
> > > W4. A uniform prior of target labels as assumed in Eq. 2 is too far from the observed reality in LiDAR object detection
> >
> >
> > **Re W4**. As discussed in **Section A.1 in the supplemental material**, in mainstream 3D detection datasets, the curated test set is commonly **long-tailed distributed**, with a few head classes (e.g., car) possessing a large number of samples and all the rest of the tail classes possessing only a few samples. As such, the trained detector can be **easily biased towards head classes** with massive training data, resulting in high accuracy on head classes and low accuracy on tail classes. This suggests that for 3D detection tasks, **mean average precision (mAP)** can be a **fairer metric** of evaluation, by taking an average of all AP values per class. **When the test label is uniformly distributed, mAP scores will be equal to the AP scores for all samples**. This motivates us to choose the uniform distribution as the prior distribution, rather than estimating the test label distribution from the initial labeled set DL. In this case, the trained model tends to be more robust and resilient to the imbalanced training data, achieving higher mAP scores.
> >
> >
> > To justify the effectiveness of choosing the uniform distribution, we provide more comparisons with the SOTA active learning methods in Table 1 and Table 2 in the supplementary material, which do not take the uniform distribution as an assumption. We clearly observe that such AL methods perform poorly on tail classes (e.g., pedestrian and cyclist), confirming that the yielded models are biased towards learning car samples.
> >
> > > W5.Stage 2 in section 2.3 is quite hard to understand. Each step in Eq. 7 seems too big a leap for me and proper discussion are not made in the text part.
> >
> >
> > **Re W5.** Thanks for the valuable feedback. The main target of Eq. 7 is to determine a subset $\mathcal{D}_{S_2}^*$ from the pre-selected set $S_1$ in the last stage, by **minimizing the set discrepancy in the latent feature space**. However, the test features are not observable during the training phase, and it is hard to guarantee that the feature distribution can be comprehensively captured. As stated in the Remark section, we **focus on the features that are not learned well from the training set** due to the zero training error assumption, and thus, reconsider the feature matching problem from a gradient perspective. In particular, we split the test set into two groups in the gradient space: (1) The previously seen test samples that can be easily recognized will cluster near the origin, (2) The novel test samples will be diversely distributed in the subspace. As the first group of samples have been sufficiently covered by the initiated, in this stage, we focus on finding matching with the latter group. By assuming the prior distribution of gradients follows a Gaussian distribution, **finding the K-metroids is naturally a choice to mitigate the gap between the mean and variance**. K-metroids algorithm breaks the dataset up into groups and attempts to minimize the distance between points labeled to be in a cluster and a point designated as the center of that cluster (*i.e.*, prototype). By selecting the prototypes in the second stage, we implicitly bridge the gap between the selected set and the test set at a latent feature level. Due to the page limit, we have included an extra paragraph of the justifications in the supplementary material (Section A.3) for a better understanding of the motivation for Stage 2.
> >
> > > W6. Only the KITTI experiments showed promising results. For the Waymo experiments in Figure 2, CRB is out-performed by LLAL and ENTROPY in the low data regime while almost on par with RAND baseline in the high data regime.
> >
> > > Q2. Why different lines in Figure 2 have different sample point w.r.t Number of Selected Bounding Boxes ?
> >
> > **Re W6 and Q2.** Thanks for your valuable comments. Regarding the first comment in W6 and Q2, we would like to clarify that, as most of AL baselines can only accept the acquisition budget at an instance level (*i.e.*, a fixed number of point clouds), we design the AL protocol (as shown in Section B.2 of the supplementary material) where in each selection round, a fixed number of point cloud (*i.e.*, N_r) will be picked to query the oracle. One instance, is the AL approach such as CORESET compares the **embeddings of point clouds** and selects the more diverse ones to form the query set. In this case, if the ROI features at the bounding box level are selected, there is no guarantee that there will be an instance around the ROI. Therefore, different AL strategies will choose different N_r point clouds, thus yielding a different number of bounding boxes to annotate and different costs of annotation time.

---

> > > ### Author Response · Authors · 2022-11-09
> > > **Initial Response to Reviewer 8qfL [Cont']**
> > >
> > >
> > > In the same vein, the methods like ENTROPY and LLAL tend to select the most difficult point clouds having high entropy and high estimated loss, which yields fewer bounding boxes to annotate on the Waymo dataset. In this case, at the extremely low budget costs (*e.g.*, **0.14%** bounding boxes / 13,000 bounding boxes in the Waymo dataset), we are comparing between the result of ENTROPY after 4 rounds of training and the one of CRB after only 1 round of training. This is a bit unfair to directly compare these two dots, since the proposed method is only trained with **$800$ point clouds for 80 epochs** while the ENTROPY and LLAL have been trained with $2000$ point clouds for 200 epochs.
> > >
> > > > W6. I find there are implementations for other datasets like nuScenes and Lyft in the supplementary codes, which makes me wonder if the method generalizes well across different datasets.
> > >
> > > **Re W6.** Our codebase is constructed based on the OpenPCDet library (https://github.com/open-mmlab/OpenPCDet), which comprises the data loaders for other datasets like nuScenes and Lyft. As our work is the first comprehensive study on active 3D detection, we re-implement and test 9 AL baselines on two benchmark datasets, and run 3 trials for each method on the KITTI dataset, which is very time-consuming. Due to the time limit, **we did not conduct experiments on nuScenes and Lyft before, and would like to add the extra experiments very soon**. We will update the comparison and add the yielded results to the revised manuscript once completed.
> > >
> > > > Q1.Why does GPDB alone in Table 2 has so wide an error bar and underperform the baseline?
> > >
> > > **Re Q1.** Thanks for raising the important question. The third criterion GPDB is a greedy search based algorithm, which randomly picks the first point clouds, then select the rest point clouds towards the objective of minimizing the KL divergence between the estimated KDE of the selected subset and the uniform distribution. This allows the trained detector to predict more accurate localization and size of bounding boxes and recognize both close (*i.e.*, dense) and distant (*i.e.*, sparse) objects at the test time. However, if the GPDB strategy is applied alone (as shown in the 4-th row of Table 2), the quality of the **first point cloud selected** will have a great influence on the subsequent greedy search, which is highly likely to trigger a **high variance**. However, if the CLS or/and RPS are applied together, the unreliable and redundant samples can be filtered in advance to stabilize the selection and also shrink the search space to speed up.
> > >
> > > ### References
> > > [1] Qi, Charles R., Hao Su, Kaichun Mo, and Leonidas J. Guibas. "Pointnet: Deep learning on point sets for 3d classification and segmentation." In Proceedings of the IEEE conference on computer vision and pattern recognition, pp. 652-660. 2017.
> > >
> > > [2] Shi, Shaoshuai, Xiaogang Wang, and Hongsheng Li. "Pointrcnn: 3d object proposal generation and detection from point cloud." In Proceedings of the IEEE/CVF conference on computer vision and pattern recognition, pp. 770-779. 2019.
> > >
> > > [3] Finlay, Chris, Jeff Calder, Bilal Abbasi, and Adam Oberman. "Lipschitz regularized deep neural networks generalize and are adversarially robust." arXiv preprint arXiv:1808.09540 (2018).
> > >
> > > [4] Shi, Shaoshuai, Chaoxu Guo, Li Jiang, Zhe Wang, Jianping Shi, Xiaogang Wang, and Hongsheng Li. "Pv-rcnn: Point-voxel feature set abstraction for 3d object detection." In Proceedings of the IEEE/CVF Conference on Computer Vision and Pattern Recognition, pp. 10529-10538. 2020.
> > >
> > > [5] Yan, Yan, Yuxing Mao, and Bo Li. "Second: Sparsely embedded convolutional detection." Sensors 18, no. 10 (2018): 3337.

---

> > > > ### Author Response · Authors · 2022-11-19
> > > > **Follow-up Re W6: Experimental Results on NuScenes Datasets**
> > > >
> > > > **Follow-up Re W6.** Below, we report the performance of different active selection strategies evaluated on the NuScenes Datasets with budget of around 65k bounding boxes in the following table.
> > > >
> > > > | |RANDOM|CORESET|LLAL|CRB|
> > > > |:----|:----|:----|:----|:----|
> > > > |NDS|33.3|32.53|32.72|**36.56**|
> > > > |mAP|21.55|21.11|20.92|**24.43**|
> > > >
> > > > Note in NuScenes dataset, there are 10 classes: 'car','truck', 'construction_vehicle', 'bus', 'trailer', 'barrier', 'motorcycle', 'bicycle', 'pedestrian', 'traffic_cone'. While in previous experiments on Waymo Open Dataset and KITTI Dataset, there were only 3 classes. The observed performance gains by the proposed CRB strategy evidences that the proposed method is capably of dealing with challenging dataset that is of long-tailed distribution. We infer that the performance gains are benefited from selecting objects with diverse and non-redundent class semantics.
> > > >
> > > > We will add these experimental results into the revised manuscript and the guidelines in our active-3D-det toolbox to support the NuScenes dataset.

---

### Official Review · Reviewer_cyaq · 2022-10-26

**Confidence:** 3
**Correctness:** 4
**Technical Novelty And Significance:** 3
**Empirical Novelty And Significance:** 3
**Recommendation:** 6

**Clarity, Quality, Novelty And Reproducibility:**

The paper is well written, easy to follow and understanding. Codes are available in the submission. The proposed method is novel to me.

**Strength And Weaknesses:**

**Strength**
1. The motivation of this paper is solid, and the proposed framework CRB is novel.
2. The paper is well written.
3. Extensive experiments are conducted to demonstrate the superiority of the proposed method.

**Weakness**
1. In Sec. 2.1, I am very confused how to annotate a point cloud $\mathcal{P}$ with a bounding box $\mathcal{B}$ and its associated box label $\mathcal{Y}$?
2. From Fig. 2 in the main paper and Fig. 5 in the Suppl., the proposed method seems to get inferior results when a low budget is required. Considering the scale of Waymo is bigger than KITTI, what is the performance on WOD under the same setting as Tab.1?
3. In B.2, there are several hyper-parameters about active learning protocols. So how to set values of them? From Tab.3, results are sensitive to the setting of $\mathcal{K}1$ and $\mathcal{K}2$, does both thresholds need to be carefully tuned when different number of annotated bounding boxes are available?


**Summary Of The Paper:**

This paper focuses on the task of 3D object detection with only a small portion of labeled data. Conventional active learning-based approaches show a promising solution, but they fail to balance the trade-off between point cloud informativeness and box-level annotation costs. To overcome this limitation, this paper jointly investigates three novel criteria for point cloud acquisition, i.e., label conciseness, feature representativeness and geometric balance. Extensive experiments are conducted on KITTI and WOD benchmarks to demonstrated the superiority of the proposed method.

**Summary Of The Review:**

This paper proposes a novel CRB framework for 3D object detection with active learning. The proposed method is novel, and achieves superior performance on both KITTI and WDO benchmarks. Although there are some questions mentioned above, I think the overall quality of this paper is good.

---

> ### Author Response · Authors · 2022-11-09
> **Initial Response to Reviewer cyaq**
>
> We thank  the reviewer for the positive comments, and important questions raised. Below, please find our response to each of the comments:
>
>
> > W1. In Sec. 2.1, I am very confused how to annotate a point cloud P with a bounding box B and its associated box label Y?
>
> **Re W1.** Thanks for your comment. As stated in [1] (Section 2.3), the general annotation process is to ask annotators to assign tractlets in the form of 3D bounding boxes to objects such as cars, pedestrians, and cyclists. A special purpose labeling tool is created, which displays 3D laser point clouds and camera images to increase the quality of annotations. For the Waymo dataset [2], the labeling process is very similar to the one used in KITTI, where each object is labeled with a 7-DOF 3D upright bounding box with a unique tracking ID (refer to Section 3.3).
>
> > W2. From Fig. 2 in the main paper and Fig. 5 in the Suppl., the proposed method seems to get inferior results when a low budget is required. Considering the scale of Waymo is bigger than KITTI, what is the performance on WOD under the same setting as Tab.1?
>
>
> **Re W2.** It is an important question. As indicated in Table 2 of [2], the Waymo dataset consists of around 12 million 3D bounding boxes, where around 80% of point clouds are used for training. This means that, with less than **0.47%** bounding boxes (i.e., 45,000), the proposed CRB strategy can achieve 52.5% of AP scores at the LEVEL 2 difficulty, while the fully supervised PV-RCNN [4,5,6,7] obtains 9% performance gains with **266.7 times more** annotation costs.
> Table 1 reports the detector performance trained with **1%** queried bounding boxes. If the detector is trained in the same setting (1% bounding boxes, 96,000) on the Waymo dataset, it is anticipated that the derived CRB will achieve higher AP and APH scores than other AL algorithms based on the trend observed in Figure 2.
> Note that, most existing AL methods can only control the budget **at an instance level**. At the fixed budget (e.g., $N_r$ = 100 for KITTI and 400 for Waymo) of selecting point clouds, for the methods like ENTROPY, they tend to select most difficult point clouds containing all uncertain objects, which yields fewer bounding boxes to annotate. In this case, at the extreme low budget costs (e.g., **0.14\%** bounding boxes / 13,000), our method has a 0.3\% lower mAP score compared with ENTROPY. However, we are comparing between the result of ENTROPY after 4 rounds of training (e.g., **200 epochs**) and the one of CRB after only 1 round of training (e.g., **80 epochs**), which is a bit unfair to the derived strategy.  We would like to emphasize that under a **reasonable** budget of annotation costs (e.g., 0.47\% for a large dataset) and sufficient training epochs, the proposed CRB achieves the state-of-the-art performance, outperforming other AL baselines.
>
> > W3. In B.2, there are several hyper-parameters about active learning protocols. So how to set values of them? From Tab.3, results are sensitive to the setting of K1 and K2, does both thresholds need to be carefully tuned when different number of annotated bounding boxes are available?
> >
> **Re W3.** Most hyperparameters we used such as the learning rate, optimizer, and LR scheduler, **follow the original setting** [3] of PV-RCNN. The batch sizes for training and test data are smaller than the original setting due to the limited computing resources. As our work is the first comprehensive study on active 3D detection task, the active training setup is empirically defined. The number of epochs $E$ and the number of rounds $R$ do not affect the model performance as long as the convergence is guaranteed. The number of point clouds to form the initiated set $m$ and the number of point clouds to query for each round $N_r$ are comparably hard to choose. With a small $m$/$N_r$, the trained detector is not reliable for acquisition, and with a large $m$/$N_r$, the computational overhead is high. We set the $m$ and $N_r$ to 2.5~3\%  point clouds (*i.e.*, 100 for KITTI, 400 for Waymo) to trade-off between reliable model training and high computational costs.
>
>
> From Table 3, We can observe that at MODERATE and HARD levels, there is **only 3.28\% and 2.81\% fluctuation on average mAP** under different settings of K1 and K2. This demonstrates a relatively stable and highly competitive performance to the full model, using only 1% bounding boxes. To support our claims that the $K_1$ and $K_2$ are not sensitive to the AL training setting, we further conduct a group of experiments on the KITTI dataset with the backbone of SECOND. The experimental results will be updated in the next reply very soon.

---

> > ### Author Response · Authors · 2022-11-09
> > **Initial Response to Reviewer cyaq - References**
> >
> > Due to the limit of characters, we include the reference list of our response below:
> >
> > ### References
> > [1] Geiger, Andreas, Philip Lenz, and Raquel Urtasun. "Are we ready for autonomous driving? the kitti vision benchmark suite." In 2012 IEEE conference on computer vision and pattern recognition, pp. 3354-3361. IEEE, 2012.
> >
> > [2] Sun, Pei, Henrik Kretzschmar, Xerxes Dotiwalla, Aurelien Chouard, Vijaysai Patnaik, Paul Tsui, James Guo et al. "Scalability in perception for autonomous driving: Waymo open dataset." In Proceedings of the IEEE/CVF conference on computer vision and pattern recognition, pp. 2446-2454. 2020.
> >
> > [3] Shi, Shaoshuai, Chaoxu Guo, Li Jiang, Zhe Wang, Jianping Shi, Xiaogang Wang, and Hongsheng Li. "Pv-rcnn: Point-voxel feature set abstraction for 3d object detection." In Proceedings of the IEEE/CVF Conference on Computer Vision and Pattern Recognition, pp. 10529-10538. 2020.
> >
> > [4] Yin, Tianwei, Xingyi Zhou, and Philipp Krahenbuhl. "Center-based 3d object detection and tracking." In Proceedings of the IEEE/CVF conference on computer vision and pattern recognition, pp. 11784-11793. 2021.
> >
> > [5] Hu, Jordan SK, Tianshu Kuai, and Steven L. Waslander. "Point density-aware voxels for lidar 3d object detection." In Proceedings of the IEEE/CVF Conference on Computer Vision and Pattern Recognition, pp. 8469-8478. 2022.
> >
> > [6] Fan, Lue, Ziqi Pang, Tianyuan Zhang, Yu-Xiong Wang, Hang Zhao, Feng Wang, Naiyan Wang, and Zhaoxiang Zhang. "Embracing single stride 3d object detector with sparse transformer." In Proceedings of the IEEE/CVF Conference on Computer Vision and Pattern Recognition, pp. 8458-8468. 2022.
> >
> > [7]Yin, Junbo, Dingfu Zhou, Liangjun Zhang, Jin Fang, Cheng-Zhong Xu, Jianbing Shen, and Wenguan Wang. "Proposalcontrast: Unsupervised pre-training for lidar-based 3D object detection." In European Conference on Computer Vision, pp. 17-33. Springer, Cham, 2022.

---

> ### Author Response · Authors · 2022-11-10
> **Follow-up Re W3: Experimental Results of Parameter Sensitivity to AL Protocols**
>
> | $ K\_1$ | $K\_2$ | Epochs ($E$) | \# of Selection ($N_r$) | EASY  | MODERATE | HARD  |
> | ------- | ------ | ------ | --------------- | ----- | -------- | ----- |
> | 5       | 3      | 40     | 100             | 78.96 | 64.27    | 59.60 |
> | 3       | 2      | 40     | 100             | 77.25 | 63.30    | 58.53 |
> | 3       | 2      | 50     | 100             | 77.68 | 64.86    | 59.74 |
> | 5       | 3      | 50     | 100             | 78.18 | 63.71    | 59.23 |
> | 3       | 2      | 40     | 200             | 78.84 | 64.56    | 59.61 |
> | 5       | 3      | 40     | 200             | 80.05 | 65.66    | 60.58 |
>
> **Follow-up Re W3.** The Table above reports the performance comparisons of three variants of AL protocols:
>
> 1) epoch 40, select 100 point clouds for each active training round (row1-row2),
> 2) epoch 50, select 100 point clouds for each active training round (row3-row4),
> 3) epoch 40, select 200 point clouds for each active training round (row5-row6).
>
> For each variant, we conducted two experiments with different values of $K_1$ and $K_2$ (*e.g.*, [5, 3], [3, 2]) to explore the parameter sensitivity of the thresholds to the active learning protocol.
> The proposed CRB method is integrated with the one-stage backbone SECOND, which is then trained and evaluated on the KITTI dataset. We report the 3D mAP metrics at all difficulty levels.
> By comparing the three different groups, we can observe that the mAP scores slightly fluctuate to different choices of $K_1$ and $K_2$ at the difficulty levels of  EASY, MODERATE and HARD, respectively:
>
> 1) 1.71%, 0.97% and 1.07%
> 2) 0.5%, 1.15%, and 0.51%
> 3)  1.21%, 1.1% and 0.97%
>
> The experimental results evidence that the hyperparameters $K_1$ and $K_2$ are not sensitive to the AL learning protocols, resulting in detection performance varying in the acceptable range from 0.5% to 1.71%.

---

### Author Response · Authors · 2022-11-09
**A Summary of Revisions**

We would like to thank all the reviewers for their time and effort providing constructive feedback. To improve the readability and clarity of the paper, we have revised both the main text and supplementary material based on the comments, which are highlighted in blue with separate tags. Building on reviewers’ comments as well as expanding on the discussion points in our responses, we have made the following changes to the manuscript:

- **Further Discussions**: we have included a paragraph of the detailed motivation of the derived representative prototype selection in Section A.3 of the supplementary material.
- **More Related Work**: we have summarized another group of recent works including AL for 3D semantic segmentation and expand the AL for 2D detection by add more relevant literatures in Section I of the supplementary material.
- **Clarifications**: we have corrected the vague sentences and expressions in Section 2.1, Remark, and Section 3.2 of the main text by adding more explanations and references.

Please let us know if there are any questions or suggestions. Thanks again for the insightful suggestions provided. We do hope our responses and the updated manuscript can help ease the concerns.

---

### Author Response · Authors · 2022-11-15
**Happy to take follow-up questions before the author-reviewer discussion closes**

Dear reviewers and AC,

Thank all for your time and insightful suggestions. As the first phase of the discussion period is about to end on Novermber 18, we are wondering whether anyone has furthur questions or comments on our revised manuscript and reponses. It is very important for us to hear the feedback from you and address the concerns before this Friday :)

Thanks again in advances!

Authors

---

### Decision · Program_Chairs · 2023-01-20

**Decision:**

Accept: notable-top-25%

**Justification For Why Not Higher Score:**

I recommended it as a spotlight. The method is good, but I think there will be more interesting papers for oral given the constraints of time. Maybe something of a more general purpose.

**Justification For Why Not Lower Score:**

The paper is good. The authors did a great work addressing the reviewers comments.

**Metareview: Summary, Strengths And Weaknesses:**

The reviewers agree that the paper is solid, novel, and well-written. They agree that the experiments are correct and show the superiority of their method. They also point out that the toolbox they have created will be helpful for future research.

There were concerns regarding notation, results of some experiments that required further details, factual errors on some claims, more details on literature review, additional experiments, specific computation cost, and questions regarding whether this method can be used indoors.

The authors carefully addressed most of the reviewer's concerns and submitted a reviewed manuscript version. Accordingly, I recommend accepting this article.

**Note From Pc:**

if the above contains the word "oral" or "spotlight" please see: "oral" presentation means -> notable-top-5% and "spotlight" means -> notable-top-25%. As stated in our emails, we are disassociating presentation type from AC recommendations